# *Staphylococcus aureus* Adaptation to the Skin in Health and Persistent/Recurrent Infections

**DOI:** 10.3390/antibiotics12101520

**Published:** 2023-10-07

**Authors:** Ana-Katharina E. Gehrke, Constanza Giai, Marisa I. Gómez

**Affiliations:** 1Centro de Estudios Biomédicos, Básicos, Aplicados y Desarrollo (CEBBAD), Departamento de Investigaciones Biomédicas y Biotecnológicas, Universidad Maimónides, Buenos Aires C1405BCK, Argentina; gehrke.ana@maimonides.edu; 2Consejo Nacional de Investigaciones Científicas y Técnicas (CONICET), Buenos Aires C1425FQB, Argentina; 3Instituto de Histología y Embriología de Mendoza, Universidad Nacional de Cuyo—(UNCuyo) CONICET, Mendoza M5502JMA, Argentina; consgiai@gmail.com; 4Facultad de Ciencias Médicas, Universidad Nacional de Cuyo, Mendoza M5502JMA, Argentina; 5Facultad de Farmacia y Bioquímica, Universidad Juan Agustín Maza, Mendoza C1006ACC, Argentina; 6Departamento de Microbiología, Parasitología e Inmunología, Facultad de Medicina, Universidad de Buenos Aires, Buenos Aires C1121A6B, Argentina

**Keywords:** *Staphylococcus aureus*, persistence, adaptation

## Abstract

*Staphylococcus aureus* is a microorganism with an incredible capability to adapt to different niches within the human body. Approximately between 20 and 30% of the population is permanently but asymptomatically colonized with *S. aureus* in the nose, and another 30% may carry *S. aureus* intermittently. It has been established that nasal colonization is a risk factor for infection in other body sites, including mild to severe skin and soft tissue infections. The skin has distinct features that make it a hostile niche for many bacteria, therefore acting as a strong barrier against invading microorganisms. Healthy skin is desiccated; it has a low pH at the surface; the upper layer is constantly shed to remove attached bacteria; and several host antimicrobial peptides are produced. However, *S. aureus* is able to overcome these defenses and colonize this microenvironment. Moreover, this bacterium can very efficiently adapt to the stressors present in the skin under pathological conditions, as it occurs in patients with atopic dermatitis or suffering chronic wounds associated with diabetes. The focus of this manuscript is to revise the current knowledge concerning how *S. aureus* adapts to such diverse skin conditions causing persistent and recurrent infections.

## 1. Introduction

*Staphylococcus aureus* is a versatile pathogen that can efficiently adapt to multiple niches, causing a wide variety of infections in humans. Approximately between 20 and 30% of the population is permanently but asymptomatically colonized in the nose, and another 30% may carry *S. aureus* intermittently [1], which is a risk factor for infection in other body sites [1,2,3,4,5]. In addition to the nose, *S. aureus* colonizes human skin as a component of the commensal flora. The skin has distinct features that make it a hostile niche for many bacteria, therefore acting as a strong barrier against invading microorganisms. Healthy skin is desiccated; it has a low pH at the surface; the upper layer is constantly shed to remove attached bacteria; and several host antimicrobial peptides are produced [6]. However, *S. aureus* is able to overcome these defenses and colonize this microenvironment. It is striking how this bacterium has the capability to adapt to very different niches and colonize not only healthy skin but also the skin under altered conditions, as is the case in individuals with atopic dermatitis or diabetic patients with foot ulcers (DFU), causing persistent and recurrent infections [7,8]. The focus of this manuscript is to revise the current knowledge regarding *S. aureus* adaptation to such diverse skin conditions.

## 2. *S. aureus* Metabolic Adaptation to the Skin

### 2.1. The Skin and Its Functions as a Cutaneous Barrier

The outermost layer of the skin, the corneal layer, is comprised of terminally differentiated keratinocytes and contains highly cross-linked keratin fibrils. Under the corneal layer are the granular, spinous, and basal layers of the epidermis. The epidermis is continuously being reformed as keratinocytes migrate from the basal layer to the corneal layer, where they are eventually shed. Below the epidermis is the dermis, which is essentially a fibrous stroma consisting of collagen and elastin fibers. There are also skin appendages such as sweat glands, sebaceous glands, and hair follicles that span these layers and open onto the skin surface. Finally, the vasculature of the skin includes a superficial and deep plexus, with additional networks around skin appendages. The superficial plexus is comprised of arterioles and venules that are interconnected by capillary loops within the papillary dermis [9].

At each of the above-mentioned layers, the skin presents different effectors that contribute to its function as a barrier. From the outside to the inside, the first barrier is determined by the microbial communities that live in the *stratum corneum*. These communities are dominated by *Actinobacteria* and gram-positive cocci species from the *Staphylococcus*, *Propionibacterium,* and *Corynebacterium* genus, and their presence prevents the establishment and proliferation of pathogenic bacteria [6,9,10]. The *stratum corneum* and epidermis conform the physical barrier, where the system of tight junctions, comprised of transmembraneous proteins that include claudins, occludin, and zona occludens, is key to maintaining the integrity of the barrier and protecting the skin against potentially invading microorganisms. The cells of the physical barrier also contribute to the chemical barrier by producing epidermal lipids. Keratinocytes deliver mainly triglycerides and cholesterol, whereas sebaceous glands secrete triglycerides, wax esters, and squalene containing sebum into the upper part of the hair follicle, thereby delivering those lipids directly onto the *stratum corneum*. Bacteria and yeasts from the microbiome then hydrolyze triglycerides into free fatty acids, contributing to the acidification of the skin [11]. The chemical barrier comprises other factors that help to maintain the moisture and the acidic pH at the skin surface; these hygroscopic compounds are aminoacids and their derivatives resulting from the proteolysis of the epidermal protein filaggrin [12]. This protein is critical for optimal skin barrier function as it binds keratin within the cytoskeleton, which is important for the formation of the *stratum corneum*, therefore minimizing water loss and preventing the entry of irritants and allergens [13]. Other components of the chemical barrier are lactate, urea, and electrolytes. The immune barrier represents the final part of the cutaneous barrier and comprises a variety of resident immune cells within the epidermis and dermis. The cellular composition of the immune barrier includes innate sentinels, such as several types of resident antigen-presenting cells, innate lymphoid cells, keratinocytes, and adaptive tissue resident memory cells, which all work together to maintain barrier integrity. These immune cells efficiently sense microbial danger signals via pathogen- and damage-associated molecular patterns (PAMPs and DAMPs) and initiate an adequate immune response, leading to subsequent tissue inflammation by the recruitment of circulating counterparts that will attempt to clear bacterial invasion. In addition to this necessary but harmful action, resident immune cells further contribute to barrier repair and homeostasis. Given that cells of the immune barrier are distributed all over the skin, this barrier is highly interconnected with other levels of the cutaneous barrier; for example, it responds to signals derived from epithelial cells and secretes molecules that orchestrate epithelial behavior [9,14]. The skin barrier in health and under pathological conditions such as atopic dermatitis (AD) or diabetic foot ulcers (DFU) and its properties are graphically represented and described in Figure 1.

### 2.2. S. aureus Colonization of Healthy Skin

The ability of *S. aureus* to persist in the skin, either as a commensal or during an infectious process, is highly dependent on its cellular structural dynamics and population heterogeneity [15]. Considering the harsh skin microenvironment, a heterogeneous population of bacteria that presents diverse subpopulations has a higher probability of surviving during the stress conditions than a homogeneous population [16,17]. *S. aureus* regulates gene expression to adapt to the niche where it replicates through global regulators and transcription factors. SigB is the alternative sigma factor that regulates general responses to stress [18]. Its levels are constant throughout the growth phase [19], but SigB can be activated under unusual pH and high osmolarity conditions present in the skin [20]. Since low pH is detrimental to *S. aureus*, the bacteria must neutralize the acidity in order to colonize the skin [21]. To counteract acidic stress, *S. aureus* down-regulates organic acid production [21,22] and up-regulates purine biosynthesis and ammonia production, increasing urease and arginine deiminase activity [23,24,25]. The genes encoding urease are highly up-regulated in response to low pH [21,22,26]. Urease converts urea to ammonia and carbon dioxide through a Ni(II)-dependent reaction [27]. It has been shown that *S. aureus* in vivo uses the ammonia produced from urea to neutralize the acidic pH, contributing to its survival and pathogenesis [24]. It has also been demonstrated that the arginine deiminase (Arc) encoded in the arginine catabolic mobile element (ACME) of the highly epidemic clone ST8 USA300 plays an important role in the adaptation of *S. aureus* to the skin. Arc produces ammonia by catabolizing arginine, which prevents the formation of nitric oxide but induces the synthesis of polyamines. The spermidine N-acetyltransferase SpeG, also encoded in ACME, confers resistance to polyamines formed as a result of arginine catabolism by Arc. Therefore, the presence of ACME in this clone is a key element that allowed the adaptation to the skin [25,28]. In addition, the upregulation of purine biosynthesis favors the development of persistent phenotypes such as those producing biofilm and being resistant to antibiotics [24,25,29].

### 2.3. S. aureus Colonization of the Skin in Individuals with Atopic Dermatitis

AD is a chronic inflammatory disease of relapsing course that is clinically characterized by periodic flares of dry, red, itchy skin lesions and pathogenically by a defective skin barrier (Figure 1), recurrent infections, and both local and systemic Th2 immune responses [30,31,32,33]. It is estimated that AD affects from 5 to 30% of children and between 2 and 10% of adults worldwide [30,34].

In AD, the pH of the skin shifts toward alkalinity, in part due to low sweat secretion and decreased levels of fatty acids. Loss of function mutations in the filaggrin gene (present in certain patients) may result in reduced levels of urocanic acid (UCA) and pyrolidone carboxylic acid (PCA) [35]. Moreover, the Th2 cytokines IL-4 and IL-13 present in the AD skin reduce filaggrin expression, and therefore the levels of UCA and PCA are decreased even in patients carrying the filaggrin wild-type gene. IL-4 and IL-13 also inhibit the production of β-defensins 2 and 3 [36,37,38]. In addition, IL-4 induces an increase in fibronectin and fibrinogen, which can promote the skin binding of *S. aureus* through fibronectin- or fibrinogen-binding proteins [39], therefore favoring bacterial colonization of the skin. Among Th2 cytokines, increased levels of IL-5, which are correlated with increased levels of IgE, have been detected in the skin of AD patients [40], and a role for this cytokine in the induction of eosinophilia has been proposed [33]. IL-31 has also been found at higher levels in lesional skin compared with non-lesional skin within the same patient [41]. The increased expression of IL-31 originates from the microbiota alteration and has an important role in pruritus, a condition that is the basis for scratching and favors the entrance of infecting microorganisms into the skin [42,43].

The particular characteristics of the skin in AD patients have been associated with alterations in the microbiota, and a reduction in microbial diversity during disease flares with an increased prevalence of *S. aureus* and *Staphylococcus epidermidis* has been demonstrated [7,44]. A meta-analysis of 95 studies showed that the prevalence of *S. aureus* carriage in AD patients was 70% on lesional skin compared with 39% in non-lesional skin or healthy control skin [45]. Moreover, the contribution of *S. aureus* to the onset, severity, and perpetuation of inflammation in the skin of AD patients has been recognized [46,47].

Clinical isolates of *S. aureus* obtained from colonized AD patients exhibit changes in gene expression and polymorphisms in metabolic genes, especially those involved in the tricarboxylic acid cycle (TCA), the fumarate-succinate axis, and the generation of terminal electron carriers [48]. Fumarate has the ability to inhibit glycolysis by binding to glyceraldehyde 3-phosphate dehydrogenase [49], a crucial component of the glycolytic pathway that converts glucose into pyruvate. Increased expression of fumarase C (encoded by *fum*C) restricts fumarate production and facilitates glycolysis, which is necessary for *S. aureus* proliferation in the skin [50]. Therefore, it appears that metabolic adaptation, driven by the reliance on glycolysis for ATP generation, represents a significant selective pressure for *S. aureus* during skin colonization. The *fum*C locus was targeted in *S. aureus* isolated from the skin of AD patients, and variants with over 100-fold induction of *fum*C expression compared with USA300 were identified [48]. Increased *fum*C expression in these isolates may lead to increased fumarate hydrolysis and prevent suppression of glycolysis.

*S. aureus* adhesion is also influenced by changes in the *stratum corneum* cell composition and morphology that occur in AD. Corneocytes expose ligands such as fibronectin, loricrin, and cytokeratin that interact with bacterial proteins as fibronectin-binding proteins A and B (FnBPA, FnBPB), clumping factor B (ClfB), and the iron-regulated surface determinant A protein (IsdA), promoting adhesion of *S. aureus* and providing resistance to antimicrobial lipids [7,51]. In healthy human skin, *S. aureus* can be found in the epidermis and within the dermis [52]. Using a mouse model of AD, increased invasion of *S. aureus* into the dermis has been observed, and this process was dependent on bacterial viability and the activity of proteases such as the V8 protease and ceramidase. Moreover, the entry of *S. aureus* through the epidermal surface led to the interaction of the bacteria with immune cells, potentiating the Th2 responses characteristic of AD [53]. This Th2 environment suppresses both the expression of filaggrin and antimicrobial peptides with functional consequences in the skin barrier, maintaining alkaline pH and decreasing antimicrobial action, therefore contributing to the loss of immune homeostasis and favoring bacterial colonization [54,55]. Moreover, the impaired skin barrier facilitates the penetration of allergens and irritants, which may result in increased sensitization to microbes and high IgE levels, which perpetuate the eczema [55].

### 2.4. S. aureus Colonization of the Skin in Individuals with DFU

In diabetic patients, foot ulcer formation is a major concern [8]. It is estimated that one in three to one in every five patients with diabetes will develop a non-healing chronic wound in their lifetime, such as a DFU, that has an alarmingly high recurrence [56,57,58,59]. The most common risk factors for these wounds to occur include diabetic neuropathy, high oxidative stress, and peripheral arterial occlusive disease. These lesions imply a breakdown in the epidermis, affecting the physical and chemical barriers of the skin (Figure 1). Diabetic wounds have unique features that make them hard to heal. Diabetic wounds exhibit deregulated angiogenesis, a chronically sustained suboptimal inflammatory response, increased levels of reactive oxygen species, and persistent bacterial colonization. In the diabetic patient, hyperglycemia has important consequences for wound healing, affecting the normal function of endothelial cells and the proliferation of keratinocytes and fibroblasts, which are essential for re-epithelialization. Moreover, hyperglycemia also leads to increased production of reactive oxygen species, which are detrimental for wound healing. There is little to no production of cathelicidin (LL37), an antimicrobial peptide known to contribute to wound healing in the skin [56,57,58,59]. In addition, the hypoxic and inflammatory environment of the DFU favors the expression of elevated levels of metalloproteinases that contribute to the important destruction of the extracellular matrix [60]. Non-healing diabetic wounds can lead to deep ulcers affecting not only the epidermis but also the dermis and hypodermis, and infection of these ulcers is a frequent complication that represents a major cause of morbidity and mortality [61]. DFU infection can result in invasive severe complications such as diabetic foot osteomyelitis (DFOM) in 10 to 15% of the patients, and *S. aureus* is the primary pathogen associated with this pathology [62].

Patients with diabetes are more frequently colonized with *S. aureus* and more susceptible to staphylococcal infections [63,64]. Although the expression of β-defensins is upregulated in DFU, this is not sufficient for microbial regulation [56,57,58,59]. Diabetic skin ulcers are often colonized by a mixed community of microorganisms, including aerobic and anaerobic bacteria as well as fungi [8]. Polymicrobial colonization is very common, but among this community, *S. aureus* is a major participant [8,65,66,67,68]. A shotgun metagenomics sequencing analysis conducted with the microbiome of DFU with no clinical signs of infection from 100 patients in the United States indicated that *Staphylococcus* was the most abundant genera (18%) and *S. aureus* was the major staphylococcal species [65]. The situation in infected DFU (grades 2–4 according to the IDSA-IWGDF scale [69]) is very similar. A study conducted with isolates from 200 diabetic individuals from India showed that half of the infected wounds were positive for *S. aureus,* and one third of those were monomicrobial. Moreover, polymicrobial infection was associated with a clinical history of amputations [70]. The high abundance of *S. aureus* in infected DFU has also been reported worldwide [67,68,71,72].

Some of the features of the microenvironment that *S. aureus* encounters in the DFU include nutritional and metabolic stress. Moreover, the defect in microvascular circulation in these patients contributes to the hypoxia that characterizes the DFU [56,57,58,59]. The available levels of oxygen will depend on the depth of the ulcer and the degree of tissue necrosis. *S. aureus* is a very versatile pathogen that, in addition to being able to promote hypoxia through tissue cytotoxicity, has multiple regulatory pathways to respire in low oxygen conditions. In response to decreased oxygen conditions, *S. aureus* upregulates genes in glycolysis, fermentation, and anaerobic respiration and represses genes of the TCA [73] through the respiratory response regulator AB (SrrAB) and the anaerobic iron-sulfur cluster-containing redox sensor regulator (AirSR) regulatory systems [74,75]. Under these conditions, *S. aureus* can use nitrate and nitrite as its final oxygen acceptors or switch to fermentative metabolism [76]. It has been described that *S. aureus* has unique carbohydrate transporters that facilitate the maximal uptake of host sugars and serve to support non-respiratory growth in inflamed tissue [77]. Moreover, *S. aureus* transcriptomic analysis using an in vivo murine model of excisional wounds revealed that genes related to numerous metabolic pathways were differentially expressed at day 3 post-inoculation in diabetic mice compared with wild-type mice, suggesting a differential adaptation in the diabetic wound microenvironment. These changes included downregulation of the lac operon, likely due to hyperglycemia [78]. In addition, in vivo studies have shown that glucose-6-phosphate (G6P) is significantly elevated and is an important metabolic signal that induces the expression of staphylococcal cytotoxins through activation of the hexose phosphate transport system, Agr, and Sae systems, causing the lysis of host neutrophils, which in turn results in severe tissue necrosis in the diabetic host [79].

## 3. Genotypic Diversity among Nasal and Skin *S. aureus* Isolates in Health and Disease

The characterization of *S. aureus* isolates by molecular typing is critical to identifying high-risk clones that are able to adapt to a certain niche. Sequence-based methods such as Pulse Field Gel Electrophoresis (PFGE), Multilocus Sequence Typing (MLST), staphylococcal protein A (*spa*) typing, SCCmec typing, and Whole Genome Sequencing (WGS) are the most commonly used to monitor the spread and circulation of the diverse *S. aureus* lineages [80]. The population structure of *S. aureus* is highly clonal, and the human strains can be grouped into a discrete number of clonal complexes (CC). The MRSA lineages are less numerous because the introduction of the SCCmec must occur into MSSA lineages that are “permissive” for this element to be acquired and maintained. Nonetheless, the molecular epidemiology data for *S. aureus* are mainly focused on MRSA, and the information available about MSSA is scarce. The predominant clones of hospital-acquired and community-acquired MRSA by geographical region have been characterized and are listed in Table 1 [80]. The MSSA population is more heterogeneous than the MRSA population. This may be related to the fact that approximately 30% of the human population carries MSSA, and its circulation precedes its emergence. Approximately 40 to 50% of MSSA isolates from different geographical areas belong to clonal complexes CC5, CC8, CC22, CC30, and CC45 (shared with MRSA), while the rest belong to lineages that contain mainly MSSA, such as CC7, CC9, CC12, CC15, CC25, CC51, and CC101. Predominant clones found among MSSA isolates from uncomplicated skin and soft tissue infections obtained in global clinical trials (including those in the USA, South America, South Africa, and Europe) are listed in Table 1 [81]. A clone that has emerged recently and represents a source of concern is MSSA ST398 (CC398), mainly with *spa* type 571, which contains the phage-encoded immune evasion cluster genes and is resistant to erythromycin. MSSA ST398 is responsible for serious infections in different geographical regions, including North America, Europe, China, and the Caribbean [82].

### 3.1. S. aureus Nasal Carriage in Healthy Individuals

Considering that nasal carriage is an important risk factor for infection [1,2,3,4,5], several studies have attempted to characterize the main clones associated with colonization of the nasal epithelium. A study conducted in the United States with a large cohort (9622 individuals of age older than 1 year as part of the National Health and Nutrition Examination Survey, 2001–2002) showed that 32.4% of the individuals were colonized with *S. aureus,* and, at that point, CC30 was the major clone found among the MSSA isolates, whereas CC5 was prevalent among MRSA [83]. A more recent study indicates that isolates belonging to CC8 are also prevalent among MRSA colonizing the nares and that this clonal complex is associated with intermittent carriage, whereas isolates belonging to CC5 are associated with persistent carriage [84]. Studies that characterized *S. aureus* isolates colonizing the noses of individuals from other regions showed that, independently of the major circulating clones, certain genotypes were prevalent among carriers. A large study conducted in the Netherlands analyzing a strain collection of non-clinical origin (*n* = 829) indicated that CC30 and CC45 accounted for almost half (47%) of all carriage isolates, with the remaining isolates belonging to CC5, CC8, CC15, CC22, and CC121 [85]. CC30 was the most prevalent lineage in healthy pediatric cohorts from Ireland [86], Scotland [87], and Korea [88]. A study including 97 participants from Saudi Arabia showed a different scenario, and high clonal diversity was described. Among the strains isolated, corresponding to 43% of the individuals tested, seventeen clonal complexes were identified, and the more frequent were CC15 (*n* = 5), CC1 (*n* = 4), CC8 (*n* = 3), CC22 (*n* = 3), CC25 (*n* = 3), and CC101 (*n* = 2) [89]. In two studies that evaluated the carriage of MRSA in children from Iran and Jordan, CC22 appears as the prevalent clonal complex, followed by CC30, CC5, and CC1 [90,91]. Studies conducted in South America showed a prevalence of CC30, CC22, CC1, ST101, and CC8 among MSSA from a student population in Chile, whereas the two MRSA isolates from the same population belonged to CC5 [92]. MRSA isolated from a population of carriers in Brazil corresponded to ST5 and ST30 [93]. Major *S. aureus* clonal complexes colonizing the nares of healthy individuals worldwide are summarized in Figure 2.

### 3.2. Lineages of S. aureus That Colonize and Infect Patients with Atopic Dermatitis

Although the prevalence of *S. aureus* in the skin colonization of AD patients has been recognized, whether certain clones are better adapted to colonize and infect the skin of these patients is still controversial. Studies conducted with AD patients, including a pediatric population from Scotland and an adult population from Denmark, indicated that MSSA isolates belonging to CC1 were prevalent [87,94]. This was in contrast with the high prevalence of CC30 among MSSA isolated from nasal samples in healthy individuals [83,87]. A previous study in Spain indicated that the most common clonal complex in isolates from AD patients (young adults) was CC5 (31.2%), followed by CC15 (18.7%), CC30 (18.7%), and CC45 (15.6%). Similarly to the study of the Scottish population, isolates from atopic individuals that had never suffered AD mainly belonged to CC30 (48.3%), but this CC was underrepresented in AD patients [95]. The low prevalence of CC30 in the skin of AD patients has also been observed in two other studies from Canada [34] and Korea [96]. In a population of 119 children and 40 adults with AD from Canada, the prevalent genotypes belonged to CC45, CC5, and CC15, followed by CC1, CC8, and CC30 [34]. In the study conducted in Korea, which included 42 patients (11–43 years old), one-third of the isolates belonged to CC1, and the second most prevalent clonal complex was CC5 [96]. A very different situation has been described in a recent study conducted in Brazil with a cohort of 106 children with AD [97]. Fifty percent of these patients have developed *S. aureus* cutaneous infections, and among these, 40% of the isolates obtained were MRSA, in contrast with what is observed worldwide regarding the association of MSSA and skin infection in AD patients [34,46,98]. In this population, CC30 was the main lineage found (34.5%), likely influenced by the high incidence of MRSA (52.2% of the MRSA were CC30) [97]. In line with these findings, the concept that the *S. aureus* population that colonizes AD patients, characterized by *spa* typing and MLST-CC, mirrors the population present in a given geographical area rather than being specific to the type of disease has also been recently proposed by Ogonowska et al. [99]. In their study, the molecular analysis of 139 *S. aureus* isolates from 80 AD patients in Poland (29 children and 51 adults) revealed that the most frequent genotype was CC7 (ST7-t091), followed by isolates belonging to CC45, CC97, and CC15, resembling general population colonization in the area. Therefore, whether certain clonal complexes are associated with skin infections in AD patients is still an open question.

Despite the descriptive studies characterizing the clones present in the skin of AD patients at certain time points that have been conducted, only a few of those have been designed to determine the putative adaptive advantages that certain clones may have to colonize and infect the skin in this population. So far, the only association reported has been the colonization by *S. aureus* belonging to CC1 and patients carrying filaggrin mutations [100]. Among host genetic conditions, mutations in the filaggrin gene have been reported as a major risk factor for the development of AD and are also associated with disease severity [13]. As impaired skin barrier function is critical in the pathogenesis of AD and might promote *S. aureus* colonization of the skin, the association between filaggrin mutations and *S. aureus* colonization has been investigated. Loss-of-function mutations such as pArg501Ter and 2282del4, present in up to 50% of Northern European AD patients but absent in Southern European patients, were not associated with the *S. aureus* colonization rate. However, *S. aureus* colonization had an impact on disease severity in these patients [101,102]. On the contrary, the single-nucleotide polymorphism (SNP) of filaggrin that is located at codon 478 (p.Pro478Ser), in spite of its low frequency worldwide, has been shown to be significantly associated with higher *S. aureus* skin colonization as well as increased disease severity [103]. A study conducted by Clausen et al. in Denmark showed that CC1 was the most frequent clonal complex among isolates from patients with mutations in filaggrin, followed by CC15 and CC45. In AD patients with wild-type filaggrin, CC15 and CC45 were found with a frequency similar to that in AD patients with filaggrin mutations, whereas CC1 was less frequent. The statistical analysis of the frequency data indicated that CC1 was found in a significantly higher number of patients with filaggrin mutations than wild-type filaggrin [100]. The potential molecular mechanism behind the association of CC1 with skin colonization in AD has been recently characterized by quantifying the in vitro adherence of AD strains to the ClfB ligand L2v, a loricrin-derived peptide [86]. AD isolates belonging to CC1 adhered very strongly to L2v. In addition, ClfB isolated from CC1 strains had significantly higher binding affinity for its ligand than ClfB from strains of other clonal complexes, including CC30. Although these differences were small, the authors propose that they could be amplified when multiple copies of ClfB are present on the surface of *S. aureus*, leading to an increase in avidity [86].

In order to further investigate the putative relationship between the carriage of certain clonal complexes and AD, the lineages present in the nose and the skin within the same individual have been characterized. Interestingly, a study conducted in Scotland showed that nasal colonization in AD patients could be attributed to strains of the CC1 lineage as opposed to the CC30 lineage found in healthy individuals [87]. Moreover, in that study, it was shown that each patient was colonized with only one CC, and colonies derived from skin and nasal sites were interspersed throughout the phylogeny, suggesting an exchange of *S. aureus* between sites rather than niche-specific populations. SNP analysis of strains from different sites indicated that the nasal carriage represented a more established population and hence, a potential source of *S. aureus* colonizing diseased skin through self-transmission. This intra-individual spread has also been proposed by Clausen et al. in a study including 101 patients in which the authors determined that 94% of the patients presenting staphylococcal colonization of the nose, lesional skin, and non-lesional skin carried the same clone in all three sites as characterized by MLST (CC) and *spa* type. Among these patients, isolates from CC1, CC15, and CC45 were predominant [94]. Similar findings were obtained by van Mierlo et al. in a population of 96 adults with AD from the Netherlands [104]. Benito et al. also observed colonization of each individual with only one CC but reported a high clonal diversity among skin isolates compared with nasal isolates in AD patients from Spain. Whereas CC30 and CC5 were predominant among nasal samples, thirteen different CCs were detected among skin isolates [98]. In a prospective cohort study in Denmark undertaken to determine the clonal dynamics of *S. aureus* colonization and infection during 1 year in 11 children with AD, samples from active eczema, anterior nose, axillae, and perineum were taken every 6 weeks. *S. aureus* colonization patterns ranged from rare colonization over transient colonization to persistent colonization by a single clone or a dynamic exchange of up to five clones. A role for household transmission was also observed by analyzing siblings [105]. Therefore, although there is evidence to suggest a correlation between the clones colonizing the nose and the lesional skin of AD patients, more studies with individuals from different geographical areas are required to achieve conclusive data.

Regarding the association between certain clones and the severity of disease, in a pediatric cohort from Spain, strains of lineages CC45 and CC5 were detected in almost all cases in AD patients with severe scoring of atopic dermatitis (SCORAD index [106]), whereas lineages CC8 and CC30 were detected in those with mild or moderate ones [98]. In these samples, strains from CC1 were underrepresented similarly to what was found in a previous study with adult AD patients from the same geographical region [95]. When the relationship between the severity of the disease and the colonizing staphylococcal genotype was evaluated in a cohort from Canada, where CC1 was abundant, it was observed that the severity of the disease was significantly higher in AD patients colonized with strains from the CC1 lineage compared with patients colonized with strains from the CC15 and CC30 lineages. The severity of the disease in AD patients colonized with strains from the CC45, CC5, and CC8 lineages was in between CC1 and CD15/CC30 [34]. In a follow-up study, Clausen et al. determined the colonization status and lineage of the strains isolated from the nose, lesional skin, and non-lesional skin 18 months and 4 years after the first screening. Fifty percent of the patients presented the same CC at follow-up and interestingly, patients that showed changes in the CC compared to the first isolates had a higher SCORAD index, indicating increased disease severity. Based on these findings, it has been proposed that changes in the colonizing lineages within individuals could be associated with disease severity and that these changes could be the cause of the flares [94]. From the studies conducted so far, whether a certain genotype is associated with disease severity in AD is still under debate. Major *S. aureus* clonal complexes identified in the skin of patients with AD worldwide are summarized in Figure 2.

### 3.3. Major Clones of S. aureus That Colonize and Infect Skin Ulcers in Diabetic Patients

The putative advantages of certain *S. aureus* clones for surviving and persisting in the very particular niche of the DFU have been evaluated by several groups. In all the studies, the *S. aureus* population colonizing or infecting the DFU was highly diverse, as determined by molecular typing methods [67,71,72,107,108]. However, certain clones were predominant in some geographical regions. A study conducted in India showed that among 30 isolates (15 from monomicrobial infections and 15 from polymicrobial infections) randomly selected from 86 patients with infected DFU, ten different clones were detected, but 50% of the isolates were within the CC1 lineage. Six isolates from polymicrobial infections were CC1 ST1, and 7 isolates from monomicrobial infections were CC1 ST772. Among these CC1 ST772 isolates, four were MRSA. Other clonal complexes reported were CC22, CC672, CC5, and CC8 [70]. A report from Algeria indicated that CC1, CC15, and CC121 were the most prevalent lineages among MSSA, whereas the Brazilian clone ST239 and the European clone ST80 accounted for 82% and 14%, respectively, of the MRSA isolates from infected DFU (grades 2–4 according to the IDSA-IWGDF scale) [71]. In Portugal, CC5 accounted for 32% of the isolates evaluated (representative of the 23 pulsotypes found among 53 isolates from infected DFU). Other CCs found were CC22, CC45, CC30, CC7, CC182, and CC8 [72]. Among these, CC45 and CC30 have been associated with severe invasive diseases [109]. A recent study conducted with patients from a Tunisian hospital, comprising individuals from different areas in Africa, showed the presence of linages CC5, CC8, CC1, and CC15 among MRSA and lineages CC1, CC12, CC22, and CC398 among MSSA. The strains belonging to CC1 were all *spa* type t127, a genotype associated with severe infections in the United States and Germany [107]. The molecular characterization of *S. aureus* isolates from patients with DFU monomicrobial infection in France showed that 18 CCs and 11 ST were found, but 38% of the isolates from patients with diabetic foot osteomyelitis (DFOM) belonged to the MSSA CC398 lineage, whereas 16% of the isolates from patients presenting only skin and soft tissue infection (SSTI) were CC45. Therefore, a significant association of these CCs with either condition (DFOM or SSTI) was observed [108]. In this study, 16% of the isolates from patients with DFOM were MRSA and belonged to the CC8, CC5, and ST22 lineages. In France there is an increasing spread of the MSSA CC398 clone. A recent study characterizing isolates obtained between 2010 and 2017 from patients with DFOM showed that the prevalence of MSSA CC398 increased from 4% in 2010 to 26% in 2017. The presence of CC398 significantly correlated with the severity of the ulcer [110]. The MSSA CC398 is also highly isolated in China and is currently spreading around the world with heterogeneous prevalence and a significant impact on severe infections such as bloodstream infections, endocarditis, and bone joint infections [107,111]. Regarding colonizing lineages, CC5 and CC8 have been described as significantly associated with uninfected DFU, which were able to heal and had a favorable outcome [112]. Major *S. aureus* clonal complexes identified in the DFU of diabetic patients worldwide are summarized in Figure 2.

In relation to antibiotic resistance, a high prevalence of MRSA is observed in infected DFU, likely due to the multiple times that these patients must attend clinical centers for wound healthcare and antibiotic treatment. In certain areas, such as India and Africa, the percentage of MRSA reaches values between 40 and 85%, with some isolates reported as multidrug-resistant (MDR) [67,70,71,107], whereas in others it is close to 10–15% [108,113]. In a study conducted with patients from the Lisbon area, most isolates from DFU were identified as *S. aureus* (77.3%), and 48.7% of them were considered MRSA [72]. Moreover, it has been reported that there is a significant association between the presence of MRSA and delayed wound healing [113].

In order to further understand the dynamics of clones within the *S. aureus* population that colonizes and infects DFU, a few studies have been designed to address the role of nasal carriage in this process. A screening conducted in 660 individuals with diabetes from Australia, which comprised initial nasal/axillary swabs with follow-up at 3–34 weeks, indicated that 40% of them were *S. aureus* carriers, and out of these, 82% were persistent carriers [114], a proportion higher than what is reported for the general population [115]. The evaluation of 79 patients with DFU without clinical signs or symptoms of infection (ulcers grade 1, according to the IDSA-IWGDF scale) showed that 32% carried *S. aureus* in the nares and 37% had colonization of the ulcer. However, only 15% had colonization of both sites, and only 7 of those 12 patients carried related strains in the nose and ulcers as determined by PFGE [113]. Other two studies, in which nasal and ulcer *S. aureus* strains were characterized, were conducted with patients that had infected DFU (grades 2–4, according to the IDSA-IWGDF scale). Out of the 236 patients evaluated in France, only 36% had *S. aureus* in the nose and the ulcer, and 65% of those patients carried the same clone in both sites. In this study, lineage ST398 was significantly associated with ulcer samples, whereas MRSA CC8 was associated with nasal samples. Other CCs found were MSSA CC30, ST398, CC15, and CC45 (in the nares) and MSSA CC15, CC45, CC8, CC30, CC5, and MRSA CC8 (in the DFU) [116]. Recently, 115 diabetic patients from Iran were screened for *S. aureus* in the nose and the infected wounds (grades 3–4, according to the IDSA-IWGDF scale). In this case, only 11% of the patients were positive for *S. aureus* at both sites, and of these, 50% carried the same clone [68]. Therefore, from the studies conducted up to the present, endogenous transmission cannot be assumed as the only source for DFU infection.

DFU colonization/infection is a chronic condition in part due to the fact that *S. aureus* can hide and persist in skin cells, thereby contributing to poor wound healing and creating a circle of wound chronicity and infection. Longitudinal studies are required to determine whether these chronic wounds are persistently infected or become re-infected multiple times. In this regard, in the only study conducted so far, 48 patients with infected DFU at the same anatomical site at each visit and with failure in wound healing were included. These patients were recruited over a period of 7 consecutive years. The follow-up was classified into three timeframes: 23 patients with a follow-up between 4 and 10 weeks, 12 patients with a follow-up between 11 and 30 weeks, and 13 patients with a follow-up beyond 30 weeks. At inclusion (48 isolates), MSSA ST30 represented the main genotype identified, followed by ST45 and CC5. MRSA was isolated only in 7 cases (CC8 and CC5). During the follow-up (62 isolates), ST398 (16%) and the Lyon clone CC8 MRSA-IV (14%) were mainly detected, followed by ST45 and ST15. During the timeframe of the study, 51% of the patients had persistent *S. aureus* colonization/infection of their DFU. Among these patients, 12 (25% of the total) presented persistent DFU colonization/infection by the same strain for a period of ≤4 weeks. However, the number of patients with identical *S. aureus* strains isolated over time significantly decreased with the increase in the follow-up period. Among the seven CCs found in the persistently infected wounds, ST15-MSSA, CC8-MRSA-IV, and CC25-MSSA were only detected in the first period of follow-up (4 to 10 weeks). ST45-MSSA was detected from the initial screening up to 30 weeks of follow-up in 2 patients. The ST22-MSSA clone was identified even after a long period (52 weeks) in one patient. Interestingly, that patient had ST30-MSSA at the initial screening and switched to ST22 by week 4 [117]. The results of this study indicate that long-term persistence of *S. aureus* in DFU is a less frequent finding compared with other chronic conditions such as lung colonization in cystic fibrosis patients [118] or patients with osteomyelitis in the long bones [119], likely due to the very hostile microenvironment of the DFU. Nonetheless, although more studies are required, certain *S. aureus* clones, such as ST45 and ST22, seemed to be able to adapt and persist in the infected wound. Interestingly, these persistent clones were MSSA, indicating that MRSA is not the only concern for the infection of chronic wounds in diabetic patients.

## 4. Regulation of Virulence Factor Expression and Its Impact on Disease Onset, Exacerbation and Chronicity

*S. aureus* is a very successful pathogen due to its capacity to efficiently sense environmental signals and rapidly adapt to changing environments. In order to do so, the majority of *S. aureus* strains encode 16 two-component signal transduction systems (TCS). Using these TCS, *S. aureus* senses a diverse array of environmental stimuli, such as nutrient concentration, cell density, pH, ionic strength, and membrane stresses [120]. TCS in *S. aureus* has been recently reviewed by Haag and Bagnoli [121]. Among the known *S. aureus* TCS, AgrCA and SaeRS have been described as major global regulators of virulence gene expression [122,123].

In the AgrCA system, the active pheromone called autoinducing peptide (AIP) is sensed by the histidine kinase (HK) AgrC, and once a threshold concentration of the AIP is reached, the response regulator AgrA becomes activated [121]. Phosphorylated AgrA is the main regulator of the *agr* autoinduction cycle and binds to the *agr* P2 and P3 promoters, leading to transcriptional activation of the *agr*ABCD operon and the regulatory small RNA called RNAIII, respectively [124]. The main effector of the *agr* quorum sensing system is RNAIII, which interacts with target mRNAs, controlling the expression of surface proteins, secreted toxins, and proteases [125]. Capsule biosynthesis and the expression of secreted proteins and toxins (i.e., lipases, proteases, nucleases, hyaluronidases, phenol-soluble modulins, α, β, γ, and δ haemolysins, leukocidins, toxic shock syndrome toxins, and exfoliative toxins) are upregulated by *agr* (recently reviewed by Haag et al. [121]). In addition, AgrA induces the expression of α- and β-phenol-soluble modulins (PSMα1-4, PSMβ1-2) by direct interaction with their respective promoters [126,127]. Expression of surface proteins such as protein A and fibronectin-binding proteins, as well as coagulase, is repressed by *agr*. Apart from its autoactivation, various environmental stimuli, such as glucose and pH changes, are known to affect *agr* expression [128].

The *S. aureus* accessory element (*sae*) TCS is another key regulator of many secreted toxins, exoenzymes, and immunomodulatory proteins critical for *S. aureus* pathogenesis. SaeR and SaeS are the response regulator and the HK of the system, respectively [122], while the other two gene products, SaeP and SaeQ, form a protein complex with SaeS to regulate the sensor kinase phosphatase activity. Thus, SaeP and SaeQ are involved in dephosphorylating activated SarR, thereby affecting the expression levels of SarR-induced genes [129]. The *sae* locus is essential for the transcription and production of α and β haemolysins (*hla*, *hlb*) and coagulase, as well as toxins of the leucocidin family [130,131,132].

*S. aureus* is able to adapt to the host microenvironment by using the TCS in coordination with several important cytoplasmic regulators, such as the SarA protein family of transcriptional regulators (SarA, Rot, and MgrA) and the alternative sigma factors (SigB and SigH) (revised by Jenul and Horswill [133]).

### 4.1. Atopic Dermatitis

Experimental data have shown a role for numerous *S. aureus* virulence factors in the pathogenesis of AD [7,134]. The carriage of genes coding for those virulence factors in strains isolated from the skin of AD patients has been investigated [96,135]. Recently, whole genome sequence analysis performed on 38 *S. aureus* strains from AD patients and healthy carriers showed a high degree of genetic heterogeneity and a shared set of virulence factors, suggesting that no genomic content is uniquely associated with AD [136]. Differential gene expression patterns rather than the acquisition/loss of virulence genes are more likely to be responsible for the onset, pathogenesis, and exacerbations of the disease. In line with this hypothesis, recent studies have focused on characterizing the activation and repression as well as the selection of mutations in global gene regulators of *S. aureus* isolated from patients with AD.

A whole genome sequencing analysis of *S. aureus* strains isolated from the skin of 268 Japanese infants 1 and 6 months after birth indicated that skin colonization by *S. aureus* at 6 months of age increased the risk of developing AD. Moreover, the presence of a functional *agr* in the colonizing strains was associated with the development of AD, whereas the acquisition of dysfunctional mutations in the *S. aureus* Agr system was primarily observed in strains from 6-month-old infants who did not develop AD. In the same study, the authors showed that the expression of a functional Agr system in *S. aureus* was required for epidermal colonization and the induction of AD-like inflammation in mice [137].

In vivo activation of the Agr quorum sensing system in the skin has been shown using a murine epicutaneous infection model. The expression of RNAIII, which is indicative of the induction of *agr*, was observed four days after epicutaneous inoculation with *S. aureus,* and the mice successfully colonized with *S. aureus* developed severe dermatitis on day 7 [138]. Moreover, the expression levels of RNAIII in wash fluid obtained from the lesional skin of patients with AD were upregulated compared with those determined in non-lesional skin from the same individuals, suggesting an important role for Agr quorum sensing in AD exacerbation [138,139,140]. The relationship between a functional *agr* and the development of AD is also supported by several studies in which the pathogenic role of toxins positively regulated by *agr* has been investigated. Among these, it has been shown that α haemolysin (also positively regulated by *sae*) induces apoptosis and necrosis of epidermal cells [135]. Moreover, this toxin induced higher proliferation of T cells and increased production of IL-31 in PMBCs from AD patients than in those from healthy individuals, highlighting the complex interaction between *S. aureus* and the host immune response in AD. A significant increase in cytokines produced by T cells (IL-2, IL-9, IL-10, and IFN-γ) and monocytes (IL-1β and TNF-α) was also observed [141]. The contribution of staphylococcal toxins to inflammation has also been demonstrated by the upregulation and release of pro-inflammatory chemokines and cytokines, including CXCL8, CCL20, TNF-α, and IL-6, in primary human keratinocytes stimulated in vitro with sublytic concentrations of synthetic PSMα3. In addition, bacterial supernatant containing α-type PSMs triggered an intense induction of pro-inflammatory mediator expression and secretion during both topical and basal layer stimulation of human skin explants, suggesting that α-type PSMs can significantly contribute to AD flares through exacerbation of skin inflammation [142]. In addition to inflammation, it has been shown that PSMα induces epidermal keratinocyte cell death and stimulates the release and secretion of the alarmins IL-1α and IL-36α. PSMα was also essential for inducing IL-17-dependent dermatitis via the release of alarmins in an epicutaneous *S. aureus* inoculation model [140]. Concomitantly with the upregulation of secreted toxins, when the Agr regulon is activated, the synthesis of protein A, a conserved cell wall protein, is decreased [143]. Due to the critical role of protein A in the modulation of the neutrophil and epithelial cell death programs in the skin [144], its low expression also contributes to tissue necrosis. In addition to acting as a transcription-modulating factor, RNAIII itself encodes PSMγ [139]. This toxin, also known as δ-toxin, can trigger mast cell degranulation, inducing Th2-type dermatitis in mice [145].

*S. aureus* strains isolated from AD patients have also been shown to produce extracellular proteolytic enzymes such as metalloproteinases and serine proteases, which can contribute to epithelial damage. The proteolytic activity of the strains from patients with AD was higher than isolates from healthy carriers, suggesting that staphylococcal proteinases may contribute to the pathogenicity of atopic dermatitis [146]. *S. aureus* also has the ability to alter the integrity of the skin barrier by subverting host responses. It has been demonstrated that this pathogen stimulates keratinocytes to increase their endogenous protease activity, including specific increases in trypsin activity and enhanced degradation of desmoglein-1 and filaggrin [147].

Studies addressing the role of SaeRS in the pathogenesis of AD are lacking. However, it has been shown that the SaeRS TCS is repressed by low pH and high NaCl concentrations [148], conditions that resemble healthy skin. On the contrary, it is activated by H_2_O_2_ and α-defensins [149], suggesting that this operon might be active in the inflamed skin of AD patients.

Although an active *agr* operon can favor the onset of the pathogenic features of AD, once the disease has been established, the maintenance of an active regulon that induces the expression of a large number of virulence factors may result in a huge metabolic burden for the bacterium. A trade-off between metabolic burden and the expression of *agr*-induced virulence factors might favor the selection of *agr*-strains. In agreement with this hypothesis, the analysis of a collection of *S. aureus* isolated from chronically infected patients with AD showed that 22% had an *agr* mutant-like phenotype [150]. Moreover, it was demonstrated that *agr* mutants of MRSA USA300 were able to persist within keratinocytes by stimulating autophagy, evading caspase-1, and inflammasome activation, reflecting the survival advantage for mutants no longer expressing *agr*-dependent toxins [150]. The presence of selective pressure favoring reduced virulence in the skin of AD patients has been demonstrated in an independent study in which the presence of two different mutations in the *agr*A gene was detected in isolates from the same patient [87].

In addition to the decrease in toxin/exoprotein production, the selection of *agr*-mutants may have other adaptive advantages for *S. aureus* persistence in the skin of chronically infected AD patients. These mutants have increased expression of adhesins such as FnBP A and FnBP B, which will favor the attachment of the bacteria to the tissue [51]. The expression of protein A is also increased in the *agr* mutants [143]. Interestingly, the characterization of a large collection of isolates from AD patients showed that more than 85% of the tested strains produced significant amounts of extracellular SpA [151]. In a recent study, it was demonstrated that membrane vesicles of *S. aureus* strains recovered from the lesional skin of AD patients had an enhanced membrane lipid and protein A content compared with the strains from the non-lesional sites and also had an enhanced proinflammatory potential [152]. Moreover, protein A has been detected in the keratinocytes as well as in the intercellular space of the epidermis of AD lesions colonized with *S. aureus* [153]. We have demonstrated that protein A has potent pro-inflammatory properties due to its capacity to trigger TNF-α-like responses mediated by the TNF-α receptor type 1 (TNFR1) in immune cells and epithelial cells [154,155]. More recently, the impact of protein A on the induction of inflammatory signaling in keratinocytes has also been determined [151,152]. TNFR1 expression has been shown to be significantly higher in immune cells from patients with AD than that in immune cells from healthy individuals, and the levels of TNFR1 expression are correlated with the severity of the disease as determined by the SCORAD index [156]. Therefore, increased expression and secretion of protein A in a microenvironment with high levels of TNFR1 may significantly contribute to perpetuating the inflammatory state in the skin during AD.

Protein A is also a component of *S. aureus* biofilm [157]. It has been shown that *S. aureus* isolates from the lesional skin of patients with AD produce a substantial amount of biofilm in vitro and are less susceptible to killing by the antimicrobial peptide LL-37 when compared with *S. aureus* grown in planktonic conditions [158]. Biofilm production is usually enhanced in the presence of a dysfunctional *agr* [139], highlighting another putative advantage for *agr*- mutants to persist in the skin. However, the impact of *agr* on biofilm-associated infection is divergent, and a functional *agr* is necessary for biofilm dispersion (via induction of PSMs with surfactant functions) [139], which explains the importance of heterogeneity in the phenotypes that form the *S. aureus* biofilm community.

Another important feature of the role of quorum sensing systems in *Staphylococcus* spp. is that members of the normal human skin microbiome, such as coagulase-negative staphylococci species (CoNS), can contribute to epithelial barrier homeostasis by producing AIPs that inhibit *S. aureus agr* and therefore toxin production. Metagenomic analysis of the AD skin microbiome has shown that the increase in the relative abundance of *S. aureus* in patients with active AD is correlated with a lower CoNS AIPs to *S. aureus* ratio, thus overcoming the capacity of these AIPs to inhibit the *S. aureus* Agr system [159]. Moreover, the strains of *S. aureus* can be organized into several groups according to their responses to the different AIPs, since each of these pheromones will only activate the *agr* response in strains belonging to the same group. Therefore, AIPs belonging to one group of *S. aureus* can inhibit activation of the *agr* response in other groups [160,161]. A study of the dynamics of *S. aureus* colonization and infection during 1 year in 11 children with AD has shown that changes in the *agr* group were associated with disease flares and a higher SCORAD index, whereas changes that implicated a different clone with the same *agr* type did not correlate with disease exacerbation [105].

The regulation of *S. aureus* virulence factor expression during atopic dermatitis and the impact on disease development are summarized in Figure 3.

### 4.2. Diabetic Foot Ulcer Infections

*S. aureus* secreted toxins significantly contribute to tissue damage and the poor healing process in diabetic ulcers [8]. However, the capacity of *S. aureus* to form biofilms, to differentiate into persister cells, including the small colony variant phenotype (SCV), and to invade and hide in skin cells is critical for its survival and persistence in this very hostile microenvironment (Figure 4).

Biofilm generation, maturation, and dissociation depend on a multitude of environmental signals and several host factors, such as different nutrients, pH, temperature, and oxygen availability, which determine the switch to this sessile growth mediated by the Agr TCS and SigB. The increased capacity to form biofilms of *S. aureus* isolated from diabetic ulcers compared with those isolated from non-diabetic patients has been demonstrated, and strains that were strong biofilm producers have been associated with grade 3 ulcers (according to the IDSA-IWGDF scale) [67,70,162,163]. Diabetic wounds are characterized by hyperglycemia and advanced glycation end products (AGEs) [164]. Glucose is the most important metabolic signal, to which *S. aureus* must respond rapidly to produce energy and cellular components related to virulence [79]. The metabolism of glucose contributes to the pathogenesis of *S. aureus* by promoting biofilm formation, resistance to NO_2_, and replication within tissues [165,166,167]. AGEs are formed by the Maillard reaction, which takes place irreversibly between amine-group compounds (proteins, lipids, and nucleic acids) and carboxides (reducing sugar groups). Unlike blood glucose, AGEs can continuously accumulate in biological tissues once formed and have been associated with diabetic complications [168,169]. AGEs have been shown to promote *S. aureus* biofilm formation in clinical isolates and laboratory strains in vitro by increasing extracellular DNA through the *S. aureus* global regulator SigB [162]. In addition, glycated proteins formed from keratin and glucose also induce biofilm formation in *S. aureus.* Therefore, glycation-inhibiting and AGE crosslink-breaking compounds are currently being assayed for their ability to inhibit biofilm production [170]. The hyperglycemic microenvironment also has an impact on the physical surface properties of the bacteria, such as hydrophobicity and surface electrical charge, increasing hydrophobic attractive force and reducing electrostatic repulsion between cells, which results in better packing of bacteria within the biofilm and more efficient retention at the host surface [171]. Using an in vivo mouse model, the effect of glucose on the enhancement of biofilm formation induced by vancomycin in *S. aureus* has also been demonstrated [165]. In addition to bacterial persistence, biofilms also contribute to the inflammatory state of chronic wounds. Human epithelial keratinocytes exposed to products secreted by *S. aureus* grown in biofilms had significantly increased levels of IL-6, IL-8, TNFα, and CXCL2 compared with those stimulated with products secreted by *S. aureus* grown planktonically [172]. Biofilm production by *S. aureus* also has a negative impact on re-epithelialization and wound healing [173,174,175].

In conditions of non-maintained pH, as it occurs in the DFU, it has been shown that the Agr system is turned off in response to glucose [128,165,166,176] which may favor the development of persister phenotypes. In this regard, the relationship between the presence of high glucose concentrations, the Agr system downregulation, biofilm formation, and SCV has been described [128,165,166,176]. Moreover, using media mimicking the hyperglycemic environment of DFU, it has been shown that *S. aureus* downregulates its secreted virulence factors and enters a state characterized by a low-virulence phenotype, the development of SCV, and the display of increased expression of genes involved in adhesion and biofilm formation [177], and the downregulation or loss of a functional *agr* has been reported to play a crucial role in this process [178,179]. The adaptive advantages that dormant phenotypes have, include reduced or no expression of virulence factors, which allows the colonization of a certain niche without clinical symptoms and also allows for antibiotic tolerance even when the bacteria lack resistance mechanisms [180]. Persister cell development is favored in the stationary phase of planktonic cultures and within biofilms, therefore contributing to the high rates of treatment failure and relapse of infection in chronic wounds [181]. Among persister cell phenotypes, SCV are *agr*- variants characterized by small colony size, slow growth, reduced metabolism, downregulated virulence genes, low cytotoxicity, and high rates of resistance to antibiotics. Changes in the microenvironment can restore the phenotype of SCV back to a virulent and fast-proliferating bacteria, reinitiating the infectious cycle [178]. In addition, SCV present a high rate of internalization in host cells. Intracellular survival within skin cells provides a protected niche where SCV can persist for a long time. The mechanisms by which *S. aureus* can persist intracellularly have been recently revised by Huitema et al. [182].

A high prevalence of SCV has been observed within *S. aureus* isolates from patients with DFOM [183], and 10% of the MRSA isolated from DFU from a cohort of 120 patients had a SCV phenotype [184]. The signals of the DFU environment that induce SCV formation and/or internalization in skin cells have not been completely elucidated yet. In order to investigate long-term global phenotypic changes that may occur in the DFU microenvironment, a wound-like medium biofilm model has been used in vitro. After 16 weeks of culture, *S. aureus* had adapted its metabolism with the development of SCV and the loss of β-hemolysin expression. A nematode model used to test the “adapted” variants suggested a decrease in virulence, which was confirmed by a significant decrease in the expression of toxin-encoding genes. Interestingly, an increased expression of genes involved in adhesion and biofilm was noted [177]. In line with these findings, a decrease in toxin expression has also been reported in persistent clones isolated from DFU patients [117].

The mechanisms underlying biofilm detachment are less understood. However, it has been established that dispersal of cells from an established biofilm requires reactivation of the Agr system [185]. Reactivation of *agr* within a biofilm has been demonstrated by Yarwood et al., and it is likely to occur through local accumulation of AIP reaching concentrations high enough to activate the Agr system [186]. Agr-dependent biofilm dispersal has been shown to occur through the action of staphylococcal proteases aureolysin and Spl [185], as well as PSMs, which disrupt the non-covalent forces holding the biofilm extracellular matrix together [187].

## 5. Conclusions and Future Perspectives

The understanding of the mechanisms behind *S. aureus* adaptation to different microenvironments has advanced considerably in the last few decades, and the capability of this microorganism to persist over long periods of time in the host is now well recognized. The development of molecular typing methods and the advancement of whole genome sequences, together with transcriptomic, proteomic, and metabolomic techniques, have provided vast information about the dynamics of *S. aureus* populations and their ability to differentially express genes according to the niche where they replicate. Important changes in metabolism through a vast array of regulatory molecules account for the ability of *S. aureus* to colonize healthy and chronically altered skin. Whereas the capacity of *S. aureus* to adapt to the nutrients available, pH conditions, and certain chemical properties of the skin have been studied, the impact that the Th2 milieu present in AD skin and the hypoxic environment of the DFU may have on the bacteria are aspects that remain to be elucidated. Whether certain genotypes are associated with recurrent and/or persistent skin infections, such as flares in AD and DFU, is still under debate. Certain strong associations, however, have been described, such as the impact of CC1 and CC45 strains on the severity of skin lesions in AD patients and the relevance of MSSA CC398 DFU infection in the ulterior development of DFOM. However, more studies are required to understand the reasons behind the apparent increased virulence of those genotypes. Moreover, data that may reveal the association of certain genotypes with increased virulence in patients with AD or DFU from more diverse geographical areas is needed. Nasal colonization seems to have an important role in AD, whereas for DFU, endogenous clones are not the only source of infection. For both types of infections, more longitudinal studies are required to better understand the dynamics of reinfections versus the persistence of a certain clone within the same patient. Altered skin conditions found in AD and DFU favor biofilm formation, which in turn can lead to the development of persistent phenotypes, including SCV, that are refractory to antibiotic treatment regardless of the presence or absence of resistance genes. Moreover, internalization of SCV within host cells may lead to long-term persistence within the host, explaining recurrent infections in these patients. The ability of *S. aureus* to develop into dormant forms helps to explain why, although MRSA have a major clinical impact, MSSA are also highly relevant in persistent infections.

## Figures and Tables

**Figure 1 antibiotics-12-01520-f001:**
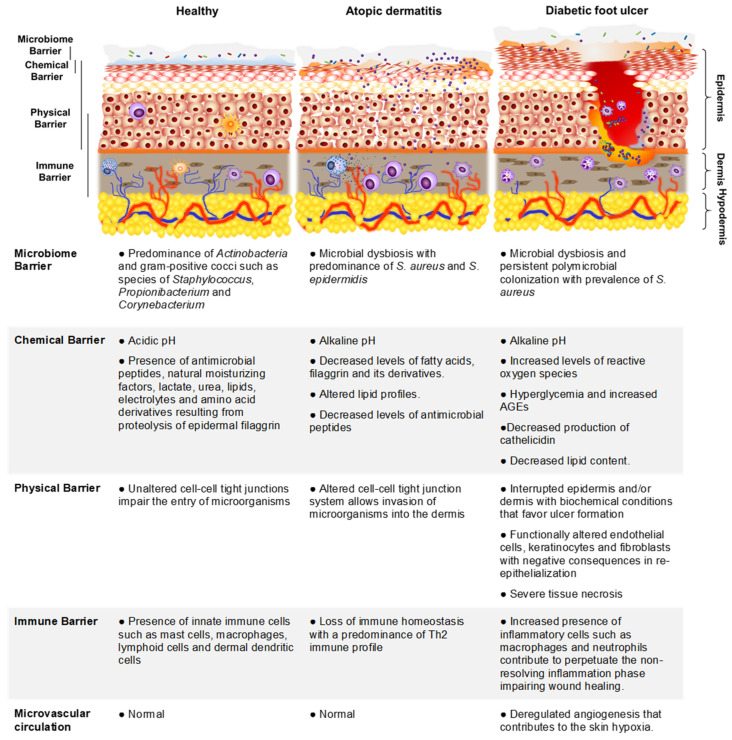
Representation of the skin-cutaneous barrier in health and disease.

**Figure 2 antibiotics-12-01520-f002:**
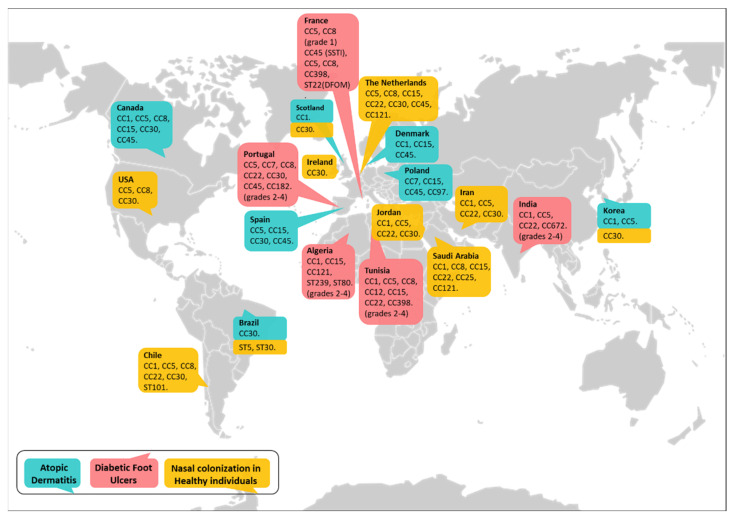
Genotypic diversity among nasal and skin *S. aureus* isolates in health and disease.

**Figure 3 antibiotics-12-01520-f003:**
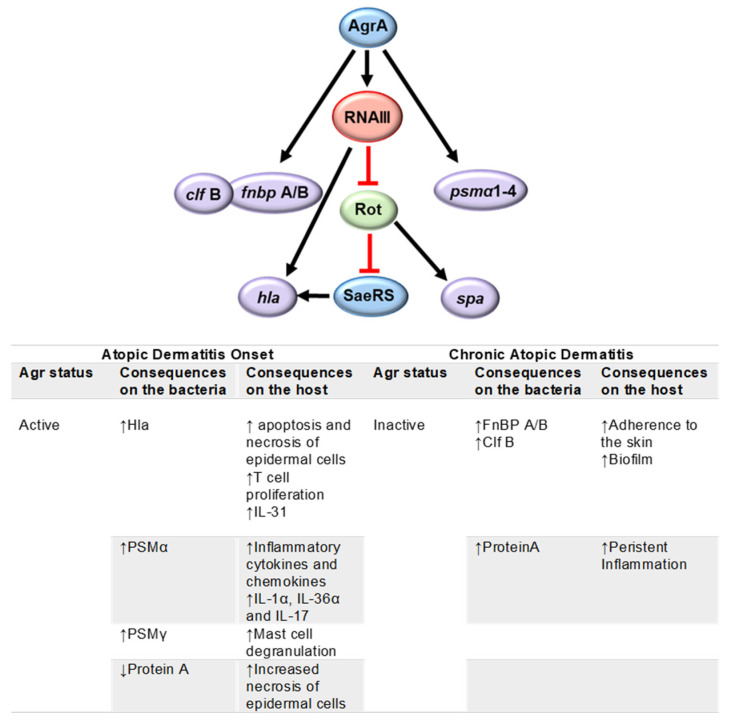
Regulation of *S. aureus* virulence factor expression in the skin during atopic dermatitis. An active Agr system is required for the initial development of atopic dermatitis. Once the disease has been established, the selection of *agr*-strains has been observed. The switch between an active and an inactive Agr system will lead to changes in the expression of virulence factors known to have a role in the pathogenesis of atopic dermatitis.

**Figure 4 antibiotics-12-01520-f004:**
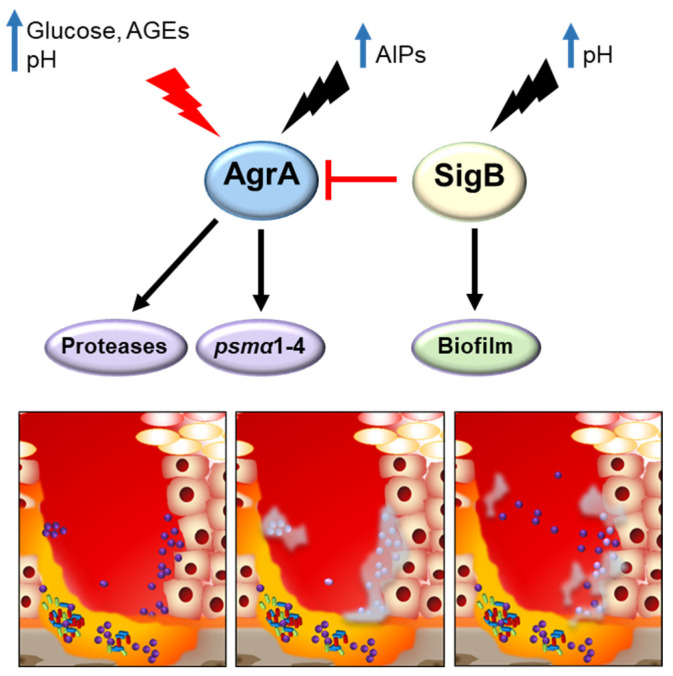
Regulation of *S. aureus* biofilm formation and detachment within the diabetic foot ulcer. Upon entry into the ulcer (**Left panel**) *S. aureus* encounters a hyperglycemic and alkaline microenvironment. The high levels of glucose under those pH conditions turn off the Agr system. Increased pH also activates SigB, which represses Agr. As a consequence of Agr downregulation and SigB activation, increased biofilm production is observed (**middle panel**). As the infection progresses, some individuals will turn back on Agr, likely due to the accumulation of AIPs and the heterogeneous conditions that the bacteria encounter within the biofilm. Agr activation leads to the expression of proteases and PSMs, which are known to be required for biofilm dispersion (**right panel**).

**Table 1 antibiotics-12-01520-t001:** Predominant *S. aureus* clones by geographical region.

Region	MRSA from Hospital Setting	MRSA from Community Setting	MSSA
**North America**	ST5, ST8, ST36, ST45	ST8, ST30	ST1, ST8, ST30, ST45, ST398 (CC398)
**South America**	ST5, ST239	ST5, ST8, ST30	ST1, ST8, ST30, ST45
**Europe**	ST5, ST8, ST22	ST8, ST30, ST80	ST1, ST8, ST30, ST45, ST398 (CC398)
**Asia**	ST5, ST22, ST239	ST8, ST30, ST59, ST88, ST93, ST772	ST398 (CC398)
**Australia and** **New Zealand**	ST1, ST22	ST5, ST93	--
**Africa**	ST5, ST239	ST80, ST88	ST1, ST8, ST30, ST45

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
