# Peer review of "Staphylococcus aureus Adaptation to the Skin in Health and Persistent/Recurrent Infections"

_antibiotics, 2023, doi:10.3390/antibiotics12101520_

Round 1
Reviewer 1 Report
Overall, the manuscript is well written and provides good description of various S. aureus colonization mechanisms. Representation of information with more figures or concise tables will make the article more readable.
Author Response
We appreciate the reviewer’s comments and suggestions that helped improve our manuscript.
Response: We agree with the reviewer and we have changed the presentation format of the information regarding the description of S. aureus prevalent clones worldwide (line 258 in the marked version). We have moved the data to Table 1 in order to facilitate the interpretation by the readers. Moreover, we have added two figures summarizing the role of S. aureus global regulators and transcription factors in the expression of virulence factors within the skin in AD (Figure 3) and in the DFU (Figure 4).
Reviewer 2 Report
This review presented by Gehrke et al. discussed thoroughly about the role of S. aureus in the pathogenesis of atopic dermatitis (AD) and diabetic foot ulcers (DFU). The cited references are current and up to date. Overall, the review is well-structured and informative, but addressing the following minor concerns would further enhance its quality.
However, I have a few minor concerns:
1) In Line 133, “Moreover, the Th2 cytokines IL-4 and IL-13 present in the AD skin…”, please note that Th2 cells not only produce IL-4 and IL-13 but also IL-5 and IL-31, which are present in AD skin and contribute to skin barrier disruption. (doi: 10.4172/2155-9899.1000110)
2) For section 2.3 (S. aureus colonization of the skin in individuals with atopic dermatitis), IL-4 produced by Th2 cells also mediates enhanced expression of fibronectin and fibrinogen, thus promoting S. aureus adhesion and later colonization could also be discussed (DOI: 10.1046/j.0022-202x.2001.01331.x).
3) It would be helpful to ensure that references are appropriately cited in certain sections, such as in Line 154-155, “…over 100-fold induction of fumC expression were identified.”; Line 613-614, “…SaeRS TCS is repressed by low pH and high NaCl concentration, conditions that resemble healthy skin.”; Line 131-132, “…Mutations in the filaggrin gen (present in cer-131 tain patients) result in reduced levels of urocanic acid”.
4) Since S. aureus plays a pivotal role in the pathogenesis of AD and DFU, it might be beneficial to include a section on potential strategies for addressing S. aureus in AD or DFU skin.
5) some typos, Line 131-132, “…filaggrin gen (present in certain patients) result in reduced levels of urocanic acid…”, genes instead of gen.
Author Response
We appreciate the reviewer’s comments and suggestions that helped improve our manuscript.
1) In Line 133, “Moreover, the Th2 cytokines IL-4 and IL-13 present in the AD skin…”, please note that Th2 cells not only produce IL-4 and IL-13 but also IL-5 and IL-31, which are present in AD skin and contribute to skin barrier disruption. (doi: 10.4172/2155-9899.1000110).
Response: We appreciate the reviewer’s comment and a sentence has been added in line 139 (marked versión) that reads as follows:
“Among Th2 cytokines, increased levels of IL-5, which correlated with increased levels of IgE, have been detected in the skin of AD patients [40] and a role for this cytokine in the induction of eosinophilia has been proposed [33]. IL-31 has also been found in higher levels in lesional skin compared with non-lesional skin within the same patient [41]. The increased expression of IL-31 originates from the microbiota alteration and has an important role in pruritus, a condition that is the base for scratching and favoring the entrance of infecting microorganisms in the skin [42,43].”
2) For section 2.3 (S. aureus colonization of the skin in individuals with atopic dermatitis), IL-4 produced by Th2 cells also mediates enhanced expression of fibronectin and fibrinogen, thus promoting S. aureus adhesion and later colonization could also be discussed (DOI: 10.1046/j.0022-202x.2001.01331.x).
Response: We appreciate the reviewer’s comment and a sentence has been added in line 136 (marked version) that reads as follows:
“In addition, IL-4 induces the increase of fibronectin and fibrinogen which can promote skin binding of S. aureus through fibronectin- or fibrinogen- binding proteins [39] therefore favoring bacterial colonization of the skin.”
3) It would be helpful to ensure that references are appropriately cited in certain sections, such as in
Line 154-155, “…over 100-fold induction of fumC expression were identified.”
Response: The sentence in line 162 (previous 154) has been revised and referenced as follows:
“The fumC locus was targeted in S. aureus isolated from the skin of AD patients and variants with over 100-fold induction of fumC expression compared with USA300 were identified [48].”
Line 613-614, “…SaeRS TCS is repressed by low pH and high NaCl concentration, conditions that resemble healthy skin.”;
Response: the appropriate reference has been added in line 625 (previous line 613).
Line 131-132, “…Mutations in the filaggrin gen (present in certain patients) result in reduced levels of urocanic acid”.
Response: The sentence in line 131 has been paraphrased and referenced as follows:
“Loss of function mutations in the filaggrin gene (present in certain patients) may result in reduced levels of urocanic acid (UCA) and pyrolidone carboxylic acid (PCA) [35].”
4) Since S. aureus plays a pivotal role in the pathogenesis of AD and DFU, it might be beneficial to include a section on potential strategies for addressing S. aureus in AD or DFU skin.
Response: we appreciate the reviewer’s comment but we consider that addressing potential strategies to manage S. aureus in AD or DFU skin, although it is a very interesting topic from the clinical point of view, it is out of the scope of the review which is focused on how the bacteria adapts to the AD and DFU skin conditions.
5) some typos, Line 131-132, “…filaggrin gen (present in certain patients) result in reduced levels of urocanic acid…”, genes instead of gen.
Response: The typo has been corrected in line 132 and the manuscript has been revised for grammar/typos.
Reviewer 3 Report
Gehrke et al present their review on Staphylococcus aureus, a very successful pathogen with high public health significance. Specifically, the review discusses, in detail, how the organism adapts to the skin in health and disease. The background provided is succinct and clearly articulated. The manuscript is well written, focused, and thorough, and its scope is carefully chosen. Especially noteworthy is the discourse on the molecular epidemiology of the pathogen in asymptomatic nasal carriage vis-à-vis disease. The authors may, however, want to highlight more clearly current gaps in our understanding regarding the organism’s adaptation to the skin.
Minor edits may be needed.
Author Response
Response: we appreciate the reviewer’s comment and we have extended the last section, “conclusions and future perspectives” to point out aspects that still require more research in order to understand the pathogenesis of staphylococcal infections in patients with AD or carrying DFU.
Reviewer 4 Report
The manuscript is well prepared and very complete, just revise in general redaction sections and whrite in italics all the microorganisms names including families and orders
Author Response
Response: We appreciate the reviewer’s comment and the text has been revised.